# Advancing Therapies for Cancer—From Mustard Gas to CAR T

**Dillon K. Jarrell [1,\*], Seth Drake [1] and Mark A. Brown [2,3,4]**

1    Department of Bioengineering, University of Colorado Anschutz Medical Campus, Aurora, CO 80045, USA; seth.drake@cuanschutz.edu
2    Cell and Molecular Biology Program, Colorado State University, Fort Collins, CO 80523, USA; mark.brown@colostate.edu
3    Department of Clinical Sciences, Colorado State University, Fort Collins, CO 80523, USA
4    Epidemiology Section, Colorado School of Public Health, Colorado State University, Fort Collins, CO 80523, USA
\*    Correspondence: dillon.jarrell@cuanschutz.edu

**Abstract:** The development of targeted therapeutics for cancer continues to receive intense research attention as laboratories and pharmaceutical companies seek to develop drugs and technologies that improve treatment efficacy and mitigate harmful side effects. In the aftermath of World War I, it was discovered that mustard gas destroys rapidly dividing cells and could be used to treat cancer. Since then, chemotherapy has remained a predominant treatment for cancer; however, the destruction of dividing cells throughout the body yields devastating side effects including off-target damage of the digestive tract, bone marrow, skin, and reproductive tract. Furthermore, the high mutation rate of cancerous cells often renders chemotherapy ineffective long-term. Therapies with improved specificity, localization, and efficacy are redefining cancer treatment. Herein, we define and summarize the principal advancements in targeted cancer treatment and briefly comment on the march towards personalized medicine in the treatment of human cancer.

**Keywords:** small molecule inhibitor; personalized medicine; precision medicine; oncology; targeted therapy; drug delivery; drug screening; chemotherapy

## 1. Introduction

Cancer remains one of the principal public health concerns throughout the world [1]. The remarkable heterogeneity of cancer, even within tissue types, has rendered it extremely difficult to diagnose and treat. Throughout most of human history, surgical resection of tumors has been the only treatment option [2]. While effective when possible, the complete removal of a tumor is notoriously difficult, and resection is impossible for metastasized tumors or blood cancers. In the early 1900s, low-dose radiation therapy emerged as a second strategy for cancer treatment. Again, while effective in some cases, it was soon discovered that radiation therapy can cause cancer as well as cure it, and is often accompanied by severe side effects [2]. Following World War I, it was discovered that mustard gas selectively kills prolific cells; soon thereafter, it underwent testing as a treatment for cancer [3,4]. The success of these experiments effectively established the third method of cancer treatment: chemotherapy. For over half a century, these three "pillars" of cancer treatment have remained largely unchanged.

While chemotherapy and radiation therapy have certainly revolutionized the clinical management of cancer, the off-target toxicity of both approaches remains a critical problem [5]. Furthermore, even the combination of these treatment strategies is often ineffective in the presence of aggressive,

metastatic, or rapidly mutating cancers. In the context of these clinical problems, two principal approaches to improving the specificity and efficacy of cancer treatments have emerged. First, advancing technologies for cancer imaging/mapping, surgical and radiation precision, or localized delivery of chemotherapeutic drugs are sharpening the three canonical pillars of cancer treatment by improving specificity (reducing side-effects) and efficacy [6,7]. Second, the identification of cancer cell-specific pathways and biochemical markers are enabling the development of targeted and personalized therapeutics [8]. These include targeted small molecule enzyme inhibitors, antibody-based therapeutics, and personalized adoptive cell transfer (ACT), such as chimeric antigen receptor T-cell (CAR T) therapy. Targeted therapy has again revolutionized cancer treatment and has become the fourth "pillar" in the clinic. Figure 1 summarizes the current strategies for cancer treatment and illustrates the advancement of the field towards powerful, personalized, patient-specific therapeutics that minimize harmful and dreaded side effects.

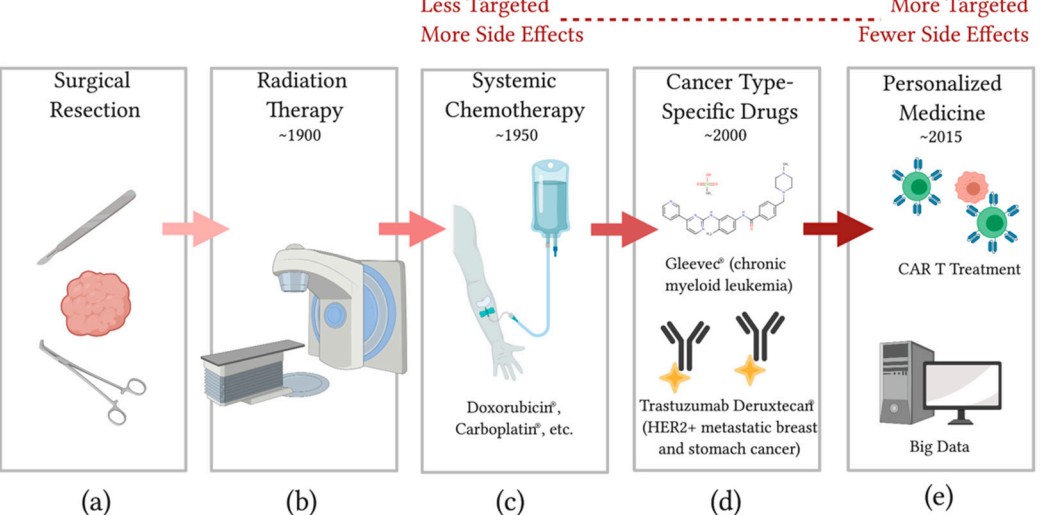

**Figure 1.** Progress of the clinical management of cancer. Throughout most of human history, surgical resection of solid tumors was the only possible treatment (**a**). Radiation therapy (**b**) can be very effective, but comes with severe side effects and, like resection, is limited to localized tumors. Chemotherapy, targeted therapies, and personalized medicine (**c**–**e**) are new, powerful drug-based strategies that aim to improve treatment efficacy and/or reduce patient side effects.

## 2. Discussion

The development and approval of Gleevec® (imatinib) in 1998, a small molecule drug that inhibits the cancer-specific BCR-ABL fusion protein in chronic myelogenous leukemias (CML), marked the dawn of targeted therapeutics [9]. Small molecule enzyme inhibitors are attractive because of their specificity, ease of manufacture, predictability, and ease of administration [10,11]. Furthermore, small molecule enzyme inhibitors can be used to target extracellular, cell-surface, and intracellular pathways, giving them a remarkable range of utility. Finally, as will be discussed below, libraries of small molecule inhibitors can be rapidly screened in high-throughput arrays, making them promising in the development of patient-specific treatment.

The recent approval of ENHERTU® (trastuzumab deruxtecan) marked the success of another approach to targeted drug delivery; the therapeutic uses a human epidermal growth factor receptor-2 (HER2)-directed antibody to deliver a topoisomerase I inhibitor (broad-spectrum chemotherapy drug) specifically to HER2+ breast cancer cells [12,13]. Rather than implementing an inhibitor specific to cancer cells, this approach uses a cancer-specific, antibody-based delivery system for a broad-spectrum chemotherapy agent. Antibody targeting of other cancer-associated pathways, including immune checkpoints, has shown great promise, and is being actively pursued in the field [14].

Within the field of targeted therapeutics, CAR T has perhaps received the most attention in recent years. CAR T involves isolating a patient's circulating killer T cells, genetically engineering the cells to express a chimeric antigen receptor that recognizes a cancer-specific antigen, and then expanding the cells and infusing them back into the patient. This "living therapeutic" trains the patient's own immune system to recognize and kill cancer cells. Third generation CAR T therapy is particularly exciting because it bypasses the need for proper MHC antigen presentation within the cancer cells, which is often defective in cancer cells due to mutation [15].

Notably, each of these three approaches to targeted therapy depend entirely on the identification of cancer-specific pathways and markers. Inhibiting the pathways or using the markers to selectively deliver cytotoxic drugs has been shown to be extremely effective, but identifying targets that are truly cancer-specific is difficult. Two approaches to this challenge in the development of targeted therapeutics are being pursued. First, as in the cases of Gleevec® and ENHERTU®, a pathway/marker that is present *predominantly* in the cancer cells of *most* patients with the specific cancer type was exploited for cancer treatment. Again, while effective, side effects are not eliminated and a significant proportion of patients do not respond. A second approach involves truly personalized medicine: using "Big Data" (the combination of patient genetics, epigenetics, transcriptomics, proteomics, RNA-seq, etc.), gene therapy, or high-throughput drug screening using patient-specific cancer biopsies, clinics can predict, develop, and prescribe "tailor-made" treatment regimens on an individual level [16–21]. This type of personalized medicine needs fine-tuning and greater validation of efficacy before it will be profitable, but is exceedingly promising. Improved identification of cancer-specific pathways and markers, streamlined development of small molecule inhibitors, effective patient data analysis, and fast high-throughput screening methods will all prove to be valuable advancements in the march towards patient-specific cancer treatment.

**Author Contributions:** Research and writing by D.K.J., Figure development by S.D., Paper conception and advising by M.A.B. All authors have read and agreed to the published version of the manuscript.

**Funding:** This research received no external funding.

**Conflicts of Interest:** The authors declare no conflict of interest.

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
