# Peer review of "Advancing Therapies for Cancer—From Mustard Gas to CAR T"

_sci, doi:10.3390/sci2040090_

Round 1
Reviewer 1 Report
This is a very well written commentary highlighting "Key Advances in Cancer Therapeutics".
Perhaps, this commentary will be strengthened if the authors could also mention targeted therapy using the checkpoint inhibitors and by including a recent review article as a supporting reference for this approach.
Author Response
This is a very well written commentary highlighting "Key Advances in Cancer Therapeutics". Perhaps, this commentary will be strengthened if the authors could also mention targeted therapy using the checkpoint inhibitors and by including a recent review article as a supporting reference for this approach. Thank you for your comments. We have indeed mentioned checkpoint inhibition and included a reference.Reviewer 2 Report
In their manuscript, the Authors summarize in brief the most pronounced advances in the systemic therapies for cancer. The manuscript is well-written and provides significant information in this area.
Minor comments:
- It would be worth mentioning that targeting the cancer-associated immune checkpoints by monoclonal antibodies has been one of the most important advances in strategies for re-shaping the specific immune response against cancer.
- Perhaps the Authors could also mention, as future perspectives, the potential change in targeted therapies against cancer that might occur if gene editing strategies enter clinical oncology with a considerable success.
Author Response
In their manuscript, the Authors summarize in brief the most pronounced advances in the systemic therapies for cancer. The manuscript is well-written and provides significant information in this area. Minor comments: 1. It would be worth mentioning that targeting the cancer-associated immune checkpoints by monoclonal antibodies has been one of the most important advances in strategies for re-shaping the specific immune response against cancer. 2. Perhaps the Authors could also mention, as future perspectives, the potential change in targeted therapies against cancer that might occur if gene editing strategies enter clinical oncology with a considerable success. Thank you for your comments. We have indeed mentioned gene therapy and antibody-immune checkpoint therapies as two important developments/future directions in the field.